# *Pneumocystis jirovecii* Pneumonia Diagnostic Approach: Real-Life Experience in a Tertiary Centre

**DOI:** 10.3390/jof9040414

**Published:** 2023-03-28

**Authors:** Cristina Veintimilla, Ana Álvarez-Uría, Pablo Martín-Rabadán, Maricela Valerio, Marina Machado, Belén Padilla, Roberto Alonso, Cristina Diez, Patricia Muñoz, Mercedes Marín

**Affiliations:** 1Department of Clinical Microbiology and Infectious Diseases, Hospital General Universitario Gregorio Marañón, 28007 Madrid, Spain; 2Instituto de Investigación Sanitaria Gregorio Marañón, 28009 Madrid, Spain; 3CIBER de Enfermedades Respiratorias (CIBERES CB06/06/0058), 08036 Madrid, Spain; 4Medicine Department, School of Medicine, Universidad Complutense de Madrid (UCM), 28040 Madrid, Spain; 5Centro de Investigación Biomédica en Red de Enfermedades Infecciosas (CIBERINFEC), Instituto de Salud Carlos III, 28220 Madrid, Spain

**Keywords:** *Pneumocystis jirovecii* pneumonia, indirect immunofluorescence, staining, polymerase chain reaction (PCR), diagnostic accuracy, immunocompromised host

## Abstract

*Pneumocystis jirovecii* pneumonia (PJP) in immunocompromised patients entails high mortality and requires adequate laboratory diagnosis. We compared the performance of a real time-PCR assay against the immunofluorescence assay (IFA) in the routine of a large microbiology laboratory. Different respiratory samples from HIV and non-HIV-infected patients were included. The retrospective analysis used data from September 2015 to April 2018, which included all samples for which a *P. jirovecii* test was requested. A total of 299 respiratory samples were tested (bronchoalveolar lavage fluid (*n* = 181), tracheal aspirate (*n* = 53) and sputum (*n* = 65)). Forty-eight (16.1%) patients fulfilled the criteria for PJP. Five positive samples (10%) had only colonization. The PCR test was found to have a sensitivity, specificity, positive predictive value (PPV) and negative predictive value (NPV) of 96%, 98%, 90% and 99%, compared to 27%, 100%, 100% and 87%, for the IFA, respectively. PJ-PCR sensitivity and specificity were >80% and >90% for all tested respiratory samples. Median cycle threshold values in definite PJP cases were 30 versus 37 in colonized cases (*p* < 0.05). Thus, the PCR assay is a robust and reliable test for the diagnosis PJP in all respiratory sample types. Ct values of ≥36 could help to exclude PJP diagnosis.

## 1. Introduction

*Pneumocystis jirovecii* is a non-cultivable ascomycete fungus responsible for a life-threatening form of pneumonia (PJP) that mostly affects immunocompromised hosts [1,2,3]. First described in malnourished children following World War II, its incidence significantly increased during the HIV epidemic [4], till highly active antiretroviral therapy and targeted prophylaxis controlled it [5]. New populations at risk have emerged, such as patients with hematological malignancies, solid organ tumors, organ transplants and connective tissue or chronic inflammatory conditions [5,6,7,8]. The mortality rates of PJP patients range from 27 to 55% [9,10,11,12,13].

Microbiological PJP diagnosis has usually been based on direct-visualization techniques with different staining methods or immunofluorescence tests. However, performance of these techniques is highly dependent on the experience and skill of the microscopist [1,2,8,9,10,11], and sensitivity may be suboptimal in patients with low fungal loads [12]. Diagnostic tests based on molecular detection of fungal DNA by PCR have superior sensitivity compared with conventional methods [6]. The usefulness of PCR for the diagnosis of PJP has been extensively studied in people living in with HIV, unlike the non-HIV population with mild or moderate immunosuppression, where clinical findings are less typical [5,7,13,14]. In addition, PCR methods can lead to over-diagnosis and over-treatment due to their inability to differentiate colonization from PJP. Therefore, positive PCR results without microscopic detection may be difficult to interpret by clinicians [11,15,16].

The aim of this study was to evaluate the performance of a real-time PCR (RT-PCR) as a routine assay in a large clinical microbiology laboratory in comparison with an immunofluorescence assay (IFA), as traditionally used in our laboratory. We also compared their performances on different respiratory samples from HIV and non-HIV-infected patients.

## 2. Materials and Methods

### 2.1. Hospital Setting, Patients and Study Population

This was an observational, retrospective study performed during a three-year period in the Hospital General Universitario Gregorio Marañón, a large, 1550-bed tertiary center, attending to a population of 350,000 inhabitants in Madrid, Spain. Ours is a referral hospital that cares for a wide variety of patients at risk of opportunistic infections, such as people living with HIV, hematopoietic stem cell or solid organ transplants recipients, rheumatologic patients and other types of immunosuppressed patients.

We included all samples with a *P. jirovecii* test requested from the clinical Microbiology laboratory with at least one respiratory sample (tracheal aspirate (TA), sputum or bronchoalveolar lavage fluid (BALF)), in which both direct immunofluorescence assay (IFA) and PCR for *P. jirovecii* (PJ-PCR) were performed blindly and simultaneously. Only one sample (the most clinically representative, classified as follows: BALF > tracheal aspirate > sputum) per patient was analyzed.

### 2.2. Microbiological Diagnosis

For microscopy testing, respiratory samples were analyzed, after concentration by cytospin cytocentrifugation, using IFA staining for cysts and trophic forms (MONOFLUO™ *Pneumocystis jirovecii* test (Bio-Rad, Marnes-la-Coquette, France)) according to the manufacturer’s instructions. All samples were reviewed and validated by the same experienced microscopist.

For molecular analyses, a real-time PCR targeting *P. jirovecii’s* large subunit of mitochondrial ribosomal DNA was performed (RealCycler PJIR kit^®^ (Progenie Molecular, Valencia, Spain)). Briefly, respiratory samples (1.5 mL) were concentrated by centrifugation at 13,000 rpm and disrupted in a MagNAlyser Instrument (Roche Diagnostics GmbH^®^, Hong Kong, China). DNA was extracted using a MagnaPure Compact instrument using the Nucleic Acid Isolation kit I (Roche Diagnostics GmbH^®^), according to the manufacturer’s instructions, with an elution volume of 50 µL. Reactions were carried out in a Smart Cycler thermal cycler (Cepheid, Sunnyvale, CA, USA). Threshold cut-off values ≤ 40 were considered as positive.

Beta-D-glucan (BDG) was detected in serum and measured using the Fungitell^®^ assay (Associates of Cape Cod, Inc., Falmouth, MA, USA), according to the manufacturer instructions. For this study, all values ≥ 80 pg/mL were interpreted as positive.

The samples were also processed for other infectious agents (bacteria, viruses, mycobacteria and fungi) according to the clinician’s request.

### 2.3. Data Collection

Demographics and clinical data were retrospectively obtained from electronic medical records according to a preestablished data collection form. Variables recorded for analysis were age, sex, main clinical symptom at the time of diagnosis and immunosuppression background (onco-hematological malignancies, HIV infection, solid organ transplant (SOT), connective tissue diseases, corticosteroid therapy). Chest imaging findings (X-ray or computed tomography (CT) scan), absolute lymphocyte count (ALC) (cell/µL), serum lactic dehydrogenase (LDH) (U/L) and value of serum BDG (pg/mL) were also collected within 7 days of the requested microbiology test, when available. Finally, *P. jirovecii* chemoprophylaxis and specific treatment were also recorded. Clinical outcome was defined either as death or as survival at 30 days after diagnosis of PJP.

### 2.4. PJP Definition

We classified a case as PJP in a patient with respiratory symptoms, suggestive radiological imaging, laboratory findings and response to therapy in the absence of an alternative diagnosis and in accordance with the principal attending physician. Patients with a positive IFA or PCR result in the absence of a suggestive clinical presentation of PJP were considered as colonized.

### 2.5. Statistical Analyses

Continuous variables were described using median and interquartile range (IQRs) and categorical variables as frequencies and percentages. Sensitivity, specificity and predictive values were calculated by the Wilson method. Association measures between categorical variables were measured by Chi^2^ and Fisher tests for 2 × 2 tables. Statistical significance was considered for *p* values < 0.05. To evaluate the diagnostic accuracy of both Ct values and the BDG threshold, we applied receiver operating characteristic methodology. The statistical analyses were performed with STATA 15,1 IC software, StataCorp LLC, TX, USA.

### 2.6. Ethics

The study was approved by the Ethics Committee of Instituto de Investigación Sanitaria Gregorio Marañón (IiSGM, Madrid, Spain) MICRO.HGUGM.2019-014, which waived the requirement of informed consent due to the design of the study.

## 3. Results

Between September 2015 and April 2018, 299 respiratory samples from an equal number of patients were simultaneously tested for *P. jirovecii* by both IFA and PJ-PCR. Samples were 181 (61%) bronchoalveolar lavage fluid samples, 53 (18%) tracheal aspirates and 65 (22%) sputa.

From the overall cohort, thirteen samples (4.3%) were IFA and PCR positive. Fifty-one samples were identified as positive PCR samples (17%), 38 of which were negative by the IFA (Figure 1). 

We identified 48 (16.1%) patients classified as PJP cases, and 251 were considered negative cases. The main characteristics of the PJP and non-PJP patients are represented in Table 1. There were not significant epidemiological differences between the groups. Among PJP cases, most patients had an underlying malignancy or HIV infection, highlighting the solid tumor group as the most frequent immunocompromised condition. Other infectious agents (bacteria, fungi, mycobacteria, and viruses) were assessed. From the 48 PJP cases, no other microorganisms were identified in 34 (70.8%) of the patients (Appendix A).

Regarding laboratory findings of patients with PJP, the median lymphocyte count was 600 cell/µL, and the median LDH value was 381 U/L. BDG-positive determination was available for 79% (*n* = 22) of the patients (median value: 623.5 pg/m; IQR: 254-1206), and diagnostic accuracy by area under curve (AUC) was 0.7 (data no shown). The highest BDG values were found in patients with solid tumors. The most common radiological patterns were interstitial (44%) and ground-glass infiltrates (31%). The mortality rate at 30 days in this group was 25% (12/48). 

### 3.1. Microbiological Diagnosis for Pneumocystis jirovecii Pneumonia

Based on the established reference definition of PJP, PJ-PCR showed higher sensitivity than IFA (95.8% vs. 27.1%). The specificity was similar for both methods (PJ-PCR 98% and IFA 100%) (Table 2).

The analysis of date based on the immunosuppression profile showed a sensitivity of 41.7% (95% CI 19.3–68) for IFA and 83.3% (95% CI 55.2–95.3) for PJ-PCR in HIV infected patients, whereas for non-HIV patients, sensitivity was 22.2% (95% CI 11.7–38.1) for IFA and 100% (95% CI 90.4–100) for PCR. The specificity for IFA and PJ-PCR was similar in both groups (100% and 98%, respectively).

Five of the fifty-one patients with a positive PJ-PCR (9.8%) were considered colonized. Ct values for these patients ranged between 32.4 to 37.6. Underlying diseases were solid tumor, solid organ transplantation, HIV, hematological malignancy, and autoimmune disease. Three patients were under anti-*P. jirovecii* prophylaxis (two with pentamidine and one with trimethoprim/sulfamethoxazole). Neither was treated for *P. jirovecii*, nor was the previous prophylaxis changed.

Negative results with both methods were obtained in 1 of the 48 patients with a definite diagnosis of PJP, and this was classified as a false negative. This patient had a prior history of HIV infection, a low CD4 count, and received Atovaquone prophylaxis with poor adherence. The medical record reported an interstitial radiological pattern accompanied by dyspnoea. Treatment for *P. jirovecii* was instituted, achieving clinical improvement. No other infectious agents were identified, nor any other non-infectious etiology of the clinical picture.

PJ-PCR showed higher sensitivity and specificity compared with IFA for all respiratory samples (Table 3).

### 3.2. Cycle Threshold PCR Stratification

Median Ct values were similar (Ct 31) for the proposed sample types. No differences were determined in Ct values among survival at the end of follow-up (Ct 30, dead or alive). On the other hand, based on the immunosuppression profile, our data showed Ct values of 25.8 in HIV infected patients and 30 for non-HIV patients. Thus, the median of Ct values was lower (30.3) in PJP patients in comparison to the colonized cases (36.6) (*p* < 0.05).

ROC estimation showed that a threshold ≥ 36 allowed differentiating colonization from real infection with a sensitivity of 80% and a specificity of 89%. The diagnostic accuracy established by the area under the curve was 0.86. Figure 2.

## 4. Discussion

*P. jirovecii* pneumonia in immunocompromised patients is a severe condition associated with a high mortality [15,17,18,19,20]. Therefore, an adequate laboratory diagnosis is key. The objective of this study was to evaluate the implementation of a real-time *P. jirovecci* PCR assay and compare the outcomes with the IFA for the diagnosis of PJP in daily practice. Our data suggest that PCR is more specific regardless of the respiratory sample, and Ct values of ≥36 could help to exclude PJP diagnosis.

Until the early 2000s, staining methods or IFA were the reference tools to achieve a diagnosis of PJP. However, the sensitivity of direct-visualization methods is deficient, particularly in non-HIV patients, probably due to a lower fungal burden [7,9]. G. W. Procop et al. described an overall sensitivity of 73.3% in four staining methods (Calcofluor white, Grocott–Gomori methenamine silver, Diff-Quik and Merifluor *Pneumocystis* stains) [21]. Moodley B. et al. reported 51% IFA positive cases in 305 patients [22]. Goterris L. et al. identified 48% of PJP cases using microscopy. There was lower sensitivity in non-HIV-infected patients compared to HIV-infected ones (44.4% and 100%, respectively) [8]. Our results are in line with the above-mentioned studies, given the sensitivity of 26% using microscopy techniques and an 87.2% NPV.

Several studies have shown an improvement in the sensitivity of *P. jirovecii* detection by PCR methods mainly in BALF or induced sputum samples. However, the increase in sensitivity carries the risk of generating positive PCR results in suspected colonized patients, sometimes making PCR results difficult to interpret clinically, because the difference between colonization and disease remains a challenge [6,23,24]. Thus, fungal-load quantification is the key to interpreting PCR results. Currently, RT-PCR tests are recommended in qualitative PCRs (nested or single round) [25,26]. Most RT-PCR tests use Ct values for interpretation of results—a relative measure of the concentration of the target in the PCR and not a real quantification of fungal load, since a standard curve using reference materials is necessary. In this sense, some authors have proposed quantitative PCRs (qPCR) using different *P. jirovecii* target genes with various Ct values, reporting sensitivity > 90%, specificity > 80% and a NPV of almost 100%, which would allow one to exclude PJP diagnosis [12,15].

In our study, a commercial RT-PCR was performed, targeting the large subunit of rRNA that shows its results in cycles. Sensitivity, specificity and NPV were 92%, 98% 98.4%, respectively, which are consistent with previous published data for other PCR assays. Other findings also highlight increased detection of PJP by PCR in both HIV and non-HIV-infected patients. This diagnostic improvement is more impressive and relevant for non-HIV-infected patients (in our series mainly represented by onco-hematology patients), given an approximately four-fold detection increase compared with IFA. Similar results were described by Doyle et. al., who suggests that increases in the detection rate are probably related to the highly sensitivity of PCR methods [6].

As for the use of C_T_ values to attempt to classify suspicious PJP cases, patients with mean Ct values of 28 were categorized as having true disease and mean Ct values of 35 and above as colonization in BALF specimens [5,12,15,27,28]. Likewise, in our data, median Ct values of 30 corresponded to true PJP cases, and median Ct values ≥ 36 were categorized as colonization (AUC 0.86, *p* value < 0.05). The few *P. jirovecii* colonization cases described in our work lead us to believe that the presence of this fungus is exceptional in patients with features suggesting PJP, or there is a trend among the attending physicians to regard such cases as a potential PJP. Some authors consider that PCR may rarely generates clinically false-positive reactions [6,15].

Ct values correspond to high and low fungal burdens and may help differentiate definitive PJP cases from colonization; however the variability of specimens may be the main obstacle to determining a cut-off value in a daily routine lab. Firstly, it is not always possible to obtain an optimal lower-respiratory-tract sample. Second, there are varying degrees of dilution in bronchoscopy, depending on the extent of the lavage. Third, respiratory samples are very heterogeneous. The European Conference on Infections in Leukemia (ECIL) guidelines recommended BALF as the reference sample to diagnose PJP [29], but several factors have to be considered, such as the patient’s medical condition and the ease of sampling. We evaluated the clinical performance of the PJ-PCR using various types of respiratory samples. The sensitivity for sputum, tracheal aspirate and BALF were 83.3%, 100% and 96.2%, respectively. The specificity values for the same samples were 91.8%, 100% and 99.4%, respectively. Our data for upper-respiratory-tract samples support the use of samples other than BALF for PJP diagnosis. Other minimally invasive detection tests have been proposed, e.g., nasopharyngeal samples and oral washings. Juliano et al., evaluated BALF and oral wash fluids with two different gene targets in 77 samples. The authors reported high sensitivity (93%) and a specificity of 56% for PJP detection [30]. Goterris et al. found similar measures of sensitivity (100%) and specificity (84.2%), and an excellent concordance (k  =  1), regardless of the targeted gene [8]. Alternatively, nasopharyngeal samples have also been used to diagnose PJP with high sensitivity (96%) and specificity (79%). They could be used to screen patients at risk of developing PJP, or for testing if there is clinical suspicion [31,32]. This approach with less invasive sampling may help optimize diagnostic strategies. However, further understanding of the operating characteristics of this promising sampling method is required.

In this era of diagnostic stewardship, molecular methods are becoming key procedures in the microbiology laboratory, and PCR has also revolutionized the field of *P. jirovecii* diagnosis. Today, PCR represents the gold standard method for this purpose. Although the commercial development of PCR has facilitated the widespread introduction of these procedures and improved both the reliability and ease of use of the technology, PCR still requires highly skilled personnel, especially for the interpretation of results, and its use it is also limited by cost. Although this approach cannot be used as a general screening method, it should be targeted at patients at high risk of developing PJP and also when clinical context requires it. On the other hand, it is worth mentioning that investing in laboratory diagnostic tools undoubtedly has a favorable impact on other costs, such as hospitalization and other diagnostic procedures, and even by avoiding unnecessary treatments.

There are some limitations to this study, especially regarding its retrospective design. Our results indicate a consistent association between BDG values (particularly with higher values) and true PJP patients, and these results are in concordance with the literature [7,33,34]. However, we cannot support these findings because complete BDG data are unavailable for the entire cohort. On the other hand, this was a real-life assessment of three distinct types of respiratory samples, which strengthens the outcomes, especially due to the inclusion of a high percentage of non-HIV immunosuppressed patients, mainly represented by onco-hematological patients.

In conclusion, PCR is a robust and reliable assay to diagnose PJP. Our results suggest that the utility of any upper respiratory samples is adequate for PJP diagnosis in routine clinical settings, without delaying the diagnosis and reducing the associated risks of invasive testing. To differentiate true infection from colonization, the incorporation of a molecular approach provided by variables such as high Ct values in PCR into the clinical diagnosis process (symptoms, radiological and laboratory findings), including the BDG assay, could be an important topic to consider. Our finding must be validated in a wider population and used for developing more accurate multivariate diagnostic models.

## Figures and Tables

**Figure 1 jof-09-00414-f001:**
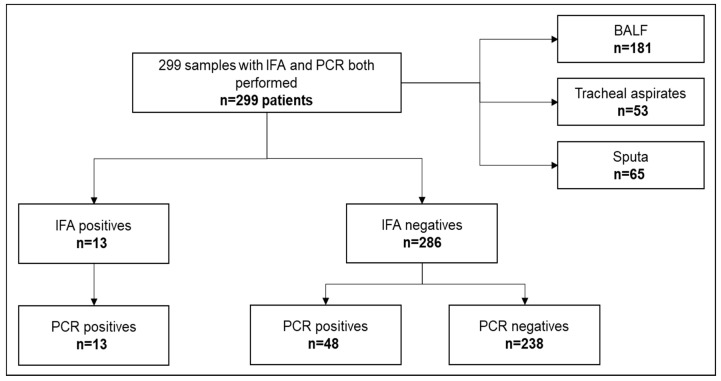
Flowchart of respiratory samples analyzed during the study period.

**Figure 2 jof-09-00414-f002:**
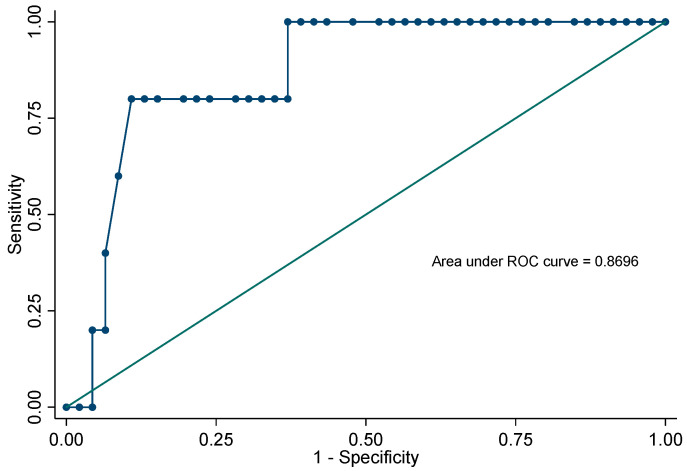
Receiver operating characteristic curve for Ct values to differentiate colonization and real infection.

**Table 1 jof-09-00414-t001:** Clinical characteristics of the patients stratified by the presence of *P. jirovecii* pneumonia.

	With PJP *n* (%)	Without PJP *n* (%)	*p* Value
Patients	48	251	
Median age (years, IQR)	57 (49–69)	58 (45–69)	0.83
Sex (male)	31 (65)	163 (65)	0.96
Underlying disease			
Hematologic malignancy	10 (21)	53 (21)	0.965
HIV infection	12 (25)	41 (16)	0.150
Solid tumor	16 (33)	36 (14)	<0.01
Solid organ transplantation	1 (2)	24 (10)	0.09
Autoimmune disease	3 (6)	16 (6)	0.97
Steroid exposure	3 (6)	9 (4)	0.39
Other ^a^	0	1 (0.4)	0.66
No immunosuppression	3 (6)	71 (28)	<0.01
Previous PJP prophylaxis	2 (4)	59 (24)	<0.01
Main clinical manifestation			
Dyspnea	29 (60)	132 (53)	0.47
Fever	12 (25)	59 (24)	0.82
Cough	7 (15)	22 (9)	0.21
No suggestive symptoms	0	38 (15)	<0.01
Radiological findings			
Interstitial	21 (44)	46 (18)	<0.01
Ground glass pattern	15 (31)	21 (8)	<0.01
Reticular	1 (2)	21 (8)	0.13
Consolidation	10 (21)	120 (48)	<0.01
Other ^b^	1 (2)	43(17)	0.02
Median ALC (cell/µL) (IQR)	600 (300–1100)	700 (300–1400)	0.29
Median LDH (U/L) (IQR)	381 (2–527)	273 (188–401)	<0.01
BDG (pg/mL) performed	28(58)	75 (30)	<0.01
<80	6 (21)	39 (52)	<0.01
≥80	22 (79)	36 (48)	<0.01
Microbiological diagnosis			
IFA positive	13 (27)	0	<0.01
PCR positive	46 (96)	5 (2)	<0.01
Median Ct (range)	30.3 (18.2–38.5)	36.6 (32.4–37.6)	<0.01
Sample collected			
BALF	25 (52)	156 (62)	0.19
Tracheal aspirate	6 (13)	47 (19)	0.30
Sputum	17 (35)	48 (19)	<0.01
Mortality	12(25)	2 (2)	<0.01

Abbreviations: PJP, *Pneumocystis jirovecii* pneumonia; PCR, polymerase chain reaction; LDH, lactate dehydrogenase; ACL, absolute lymphocyte count; BDG, beta-D glucan; IFA, immunofluorescence assay; Ct, cycle threshold; ^a^ Other, common variable immunodeficiency; ^b^ Other, pleural effusion, tumor, and atelectasis, not clear infiltrate; BALF, bronchoalveolar lavage fluid.

**Table 2 jof-09-00414-t002:** Comparison of IFA and PJP PCR assays.

	IFA 95% CI	PCR 95% CI
Sensitivity	27.1% (16.6–41)	95.8% (86–98.8)
Specificity	100% (98.5–100)	98% (95.4–99.1)
PPV	100% (77.2–100)	90.2% (79.0–95.7)
NPV	87.2% (83.5–91.1)	99.2% (97.1–99.8)

Abbreviations: IFA, immunofluorescence assay; CI, confidence interval; PPV, positive predictive value; NPV, negative predictive value.

**Table 3 jof-09-00414-t003:** Validation of PCR and IFA assays by type of respiratory sample.

	BALF % (95% CI)		Tracheal Aspirate % (95% CI)		Sputum % (95% CI)	
	IFA	PCR	IFA	PCR	IFA	PCR
Sensitivity	24 (11.5–43.4)	100 (86.7–100)	33.3 (9.7–70)	100 (61–100)	29.4 (13.3–53.1)	83.2 (65.7–96.7)
Specificity	100 (97.6–100)	99.4 (96.5–100)	100 (92.4–100)	100 (92.4–100)	100 (92.6–100)	91.7 (80.4–97)
PPV	100 (61–100)	96.2 (81.1–99.3)	100 (34.2–100)	100 (61–100)	100 (56.6–100)	78.9 (56.7–91.5)
NPV	89.1 (83.7–92.9)	100 (96.9–100)	92.2 (81.5–97)	100 (92.4–100)	80 (68.2–88.2)	95.7 (85.5–98.8)

Abbreviations: BALF, bronchoalveolar lavage fluid; CI, confidence interval; PPV, positive predictive value; NPV, negative predictive value.

## Data Availability

Not applicable.

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
