# Peer review of "Pneumocystis jirovecii Pneumonia Diagnostic Approach: Real-Life Experience in a Tertiary Centre"

_jof, 2023, doi:10.3390/jof9040414_

Round 1

Reviewer 1 Report

This manuscript shows an interesting approach on the search for a more specific and sensitive diagnostic method for PJP. A diagnostic test evaluation should include a comparison with the gold standard, which now are the molecular methods for Pneumocystis. The authors focused only on comparing one technique versus the other, although there is still discussion on which is the best molecular marker, and many authors encourage the use of the ITS gene. The authors should describe their motives for using the mithocondrial subunit DNA instead of the ITS gene.

Also, the clinical definition of PJP is weak, considering that other microorganisms might have caused the clinical symptoms of pneumonia, over a colonizing Pneumocystis. It would be interesting that the authors showed the data about the other microorganisms and if they were recovered from cultures or detected by other methods. It may be acceptable as a supplementary figure or table, considering it is not the objective of this paper, but this information would state a noteworthy conclusion about the use of RT-PCR for Pneumocystis diagnosis. 

Section 2.1

The term “HIV positive” is not adequate anymore. International AIDS associations refer to patients as “people living with HIV”; “patients with HIV infection” is also tolerated.

I suggest modifying the phrase: “hematopoietic stem cell or solid organ transplant recipients…”

How did you decide which sample was “the most clinically representative”?

Threshold cut-off values ≤40 were considered as positive

Table 1:

What does “non evaluable” mean in the Radiologic Findings section?

Did patients present with more than one symptom? i.e. dyspnea and cough

Discussion:

Some paragraphs are difficult to understand; please check English edition.

Regarding the use of nasopharyngeal swabs for diagnosis of pneumonia, there is still a great dilemma that the authors should address in their discussion: Does finding a microorganism (Pneumocystis in this case) in the nasopharynx means that it is also in the lung causing pneumonia? Microbiome studies have revealed that there are different groups of microorganisms in each part of the respiratory tract, and they can also be found in different concentrations depending on the clinical sample.

Considering the discussion on the Ct values, the authors should mention that quantitative RT-PCR is entirely different from qualitative RT-PCR, and that the Ct value itself cannot be directly interpreted as fungal load without a standard curve using reference materials. In this matter, the authors should discuss their interpretation of the Ct value considering the use of single-copy genes or methods for relative quantification of qPCR data.

Author Response

Dear reviewer of Journal of Fungi,

We appreciate your feedback in response to Brief Report titled “Pneumocystis jirovecii pneumonia diagnosis approach: Real-life experience from a tertiary center” (Manuscript ID: jof-2227896). We also recognize the great effort of the reviewers to judge our work and we appreciate your comments and suggestions.

Following the reviewer's recommendation, we have revised our manuscript and incorporate their suggestions in a tracked version to better address the peer review and we have also prepared a clean version (both attached in submission website). Additionally, we performed a deep review on language spelling by a native speaker.

Please, find our detailed answers herein to the reviewers’ queries, which have definitely increased the value of our article. We have addressed all of them carefully and included the needed changes in the manuscript.

We hope that we have adequately covered all the points suggested and we can move further in the publication process.

Thank you again to consider our team to share our work in this field. We look forward to your final decision.

Sincerely,

Cristina Veintimilla, MD

Clinical Microbiologist

Reviewer 2 Report

In this single center retrospective study performed at a clinical microbiology laboratory of a large tertiary medical center in Spain, the study authors evaluated the performance characteristics of IFA versus PCR testing for Pneumocystis jiroveci pneumonia. They noted that PCR testing had a higher sensitivity, with similar specificity, positive and negative predictive values.

Overall, the study is well designed and helps put forward more data on the utility of PCR testing in the PJP diagnosis. I just had a few comments as below.

Major Comments

1. The study authors note that when they looked at the subgroup of immunosuppressed patients, PCR testing had a higher sensitivity in both HIV and non-HIV patients. Is the data related to specificity also available for these two sub-groups of patients? This is particularly important, given the disproportionate occurrence of PJP in immunosuppressed patient populations.

2. One important factor, in this era of diagnostic stewardship, that I think needs to be addressed is the potential cost of the PCR testing. That is usually a limiting step for widespread implementation of these testing parameters and so it would also necessitate using PCR testing for the most high-risk patients. It would be worthwhile if the authors could comment or discuss this particular point, to raise awareness amongst the readers.

Minor Comments

1. Abstract – instead of “an immunofluorescence assay (IFA) in the routine”, consider changing to “routinely performed immunofluorescence assay (IFA)”.

2. Discussion (Page 8/11)– please consider changing “Despite of our data” to “Although our data”.

Author Response

(The authors gave the same response as above.)

Round 2

Reviewer 1 Report

I agree with the changes and extra information added by the authors, so I believe the manuscript is now ready for publication.